# Inhalative as well as Intravenous Administration of H_2_S Provides Neuroprotection after Ischemia and Reperfusion Injury in the Rats’ Retina

**DOI:** 10.3390/ijms23105519

**Published:** 2022-05-15

**Authors:** Stefanie Scheid, Max Goeller, Wolfgang Baar, Jakob Wollborn, Hartmut Buerkle, Günther Schlunck, Wolf Lagrèze, Ulrich Goebel, Felix Ulbrich

**Affiliations:** 1Department of Anesthesiology and Critical Care, Medical Center-University of Freiburg, Faculty of Medicine, University of Freiburg, 79106 Freiburg, Germany; stefanie.scheid@uniklinik-freiburg.de (S.S.); max.goeller@yahoo.de (M.G.); wolfgang.baar@uniklinik-freiburg.de (W.B.); jwollborn@bwh.harvard.edu (J.W.); hartmut.buerkle@uniklinik-freiburg.de (H.B.); 2Department of Anesthesiology, Perioperative and Pain Medicine, Brigham and Woman’s Hospital, Harvard Medical School, Boston, MA 02115, USA; 3Eye-Center, Medical Center-University of Freiburg, Faculty of Medicine, University of Freiburg, 79106 Freiburg, Germany; guenther.schlunck@uniklinik-freiburg.de (G.S.); wolf.lagreze@uniklinik-freiburg.de (W.L.); 4Department of Anesthesiology and Critical Care Medicine, St. Franziskus-Hospital, 48145 Muenster, Germany; ulrich.goebel@sfh-muenster.de

**Keywords:** hydrogen sulfide, H_2_S, GYY 4137, ischemia-reperfusion injury, apoptosis, inflammation, neuroprotection, retinal ganglion cell

## Abstract

Background: Neuronal ischemia-reperfusion injury (IRI), such as it can occur in glaucoma or strokes, is associated with neuronal cell death and irreversible loss of function of the affected tissue. Hydrogen sulfide (H_2_S) is considered a potentially neuroprotective substance, but the most effective route of application and the underlying mechanism remain to be determined. Methods: Ischemia-reperfusion injury was induced in rats by a temporary increase in intraocular pressure (1 h). H_2_S was then applied by inhalation (80 ppm at 0, 1.5, and 3 h after reperfusion) or by intravenous administration of the slow-releasing H_2_S donor GYY 4137. After 24 h, the retinas were harvested for Western blotting, qPCR, and immunohistochemical staining. Retinal ganglion cell survival was evaluated 7 days after ischemia. Results: Both inhalative and intravenously delivered H_2_S reduced retinal ganglion cell death with a better result from inhalative application. H_2_S inhalation for 1.5 h, as well as GYY 4137 treatment, increased p38 phosphorylation. Both forms of application enhanced the extracellular signal-regulated kinase 1/2 (ERK1/2) phosphorylation, and inhalation showed a significant increase at all three time points. H_2_S treatment also reduced apoptotic and inflammatory markers, such as caspase-3, intracellular adhesion molecule 1 (ICAM-1), vascular endothelial growth factor (VEGF), and inducible nitric oxide synthase (iNOS). The protective effect of H_2_S was partly abolished by the ERK1/2 inhibitor PD98059. Inhalative H_2_S also reduced the heat shock response including heme oxygenase (HO-1) and heat shock protein 70 (HSP-70) and the expression of radical scavengers such as superoxide dismutases (SOD1, SOD2) and catalase. Conclusion: Hydrogen sulfide acts, at least in part, via the mitogen-activated protein kinase (MAPK) ERK1/2 to reduce apoptosis and inflammation. Both inhalative H_2_S and intravenous GYY 4137 administrations can improve neuronal cell survival.

## 1. Introduction

Ischemia-reperfusion injury (IRI) plays a critical role in various disease pathologies. In the central nervous system (CNS), ischemia occurring after a stroke or traumatic brain injury leads to neuronal cell death [1]. Neuronal cell death is also seen in various ophthalmologic diseases, such as IRI of the eye secondary to glaucoma, retinal vascular occlusion, diabetic retinopathy, and retinopathy of prematurity [2,3]. Retinal ganglion cells (RGCs) are particularly susceptible to ischemia [3]. At the molecular level, IRI activates several signaling pathways resulting in increased oxidative stress, inflammation, and programmed cell death [4]. However, the CNS, including the retina, is characterized by a limited regeneration potential; therefore, affected individuals often show extended disabilities after IRI [2]. Consequently, treatments that can minimize IRI-induced neuronal cell death are highly desirable. One promising therapy in this context is the exposure to hydrogen sulfide (H_2_S).

Traditionally regarded as a poisonous gas, H_2_S has recently been recognized as an important signaling molecule with a variety of physiological functions across several organ systems, including the cardiovascular, respiratory, and central nervous systems [5]. As with nitric oxide (NO) and carbon monoxide (CO), it is classified as an endogenous gasotransmitter [6]. H_2_S readily diffuses across lipid membranes and may modulate various biological pathways at physiological concentrations [7]. By contrast, exposure to unphysiologically high concentrations of H_2_S can result in neurotoxicity [8]. In the CNS, the synthesis of H_2_S seems to rely predominantly on the enzyme cystathionine ß-synthase [9], which is expressed by astrocytes and, to a lesser extent, by microglial cells and neurons [10,11]. In the brain H_2_S is stored mainly as bound sulfur and is released upon excitation [12].

Previous studies have described the potential therapeutic value of H_2_S as a treatment for various retinal diseases [13]. The neuroprotective effect of H_2_S has been attributed to its ability to inhibit glial cell activation thus reducing inflammation [14], apoptosis [15], and oxidative stress [16]. Biermann et al. showed an anti-apoptotic effect in retinal IRI in rats following rapid pre-conditioning with 80-ppm-inhaled H_2_S and proposed a potential differential expression of mitogen-activated protein kinases as an underlying mechanism [17]. A recent study by our own group has shown both time- and dose-dependent benefits of inhalative post-conditioning with H_2_S on RGC survival after ischemia-reperfusion injury [18]. H_2_S reduced pro-inflammatory cytokine expression and NF-kB phosphorylation. However, the underlying molecular mechanisms are not yet fully elucidated and further research is needed to establish the optimal dosage, time, and mode of application.

In the present study, we addressed these questions by conducting experiments in a rat model of retinal IRI to determine the effects of either H_2_S inhalation or intravenous application of the H_2_S donor GYY 4137 on several intracellular signaling molecules that are key components of inflammatory and apoptotic pathways.

## 2. Results

### 2.1. The Protective Effect of H2S Is More Pronounced with Inhalation Therapy Than Intravenous Administration

First, we examined the effect of H_2_S on cell death and analyzed the RGC count after unilateral IRI. As expected, IRI reduced the RGC count by 40% (Figure 1, Col. 1 and 2, untreated: 2751 ± 99 vs. IRI: 1445 ± 143, *p* < 0.001) and H_2_S therapy significantly increased RGC survival after IRI. H_2_S inhalation demonstrated a protective effect both after immediate application and especially after a delayed administration of 1.5 h after IRI (Col. 2, 3, and 4, IRI: 1445 ± 143 vs. IRI + 80 ppm H_2_S at 0 h: 1879 ± 150, vs. IRI + 80 ppm H_2_S at 1.5 h: 2253 ± 177, both *p* < 0.001). Further delay had no influence on RGC. The neuroprotective effect after inhalation (1.5 h) was even more pronounced than after GYY 4137 treatment (Col. 4 and 6, IRI: 1445 ± 143 vs. IRI + 80 ppm H_2_S at 1.5 h: 2253 ± 177 and IRI + GYY 4137: 2039 ± 179, *p* < 0.001).

### 2.2. H_2_S Increases Retinal p38 and ERK-1/2 Phosphorylation after Ischemia-Reperfusion Injury

We investigated the role of mitogen-activated protein kinases after IRI and H_2_S post-conditioning by analyzing protein phosphorylation. Inhalation of 80 ppm H_2_S significantly increased p38 phosphorylation at 1.5 h after IRI (Figure 2a, Col 3, p-p38—IRI: 1.07 ± 0.23 vs. IRI + 80 ppm H_2_S at 1.5 h: 1.43 ± 0.16, *p* < 0.01), while 80 ppm H_2_S inhalation directly after IRI did not increase p38 expression. In contrast, the effect of H_2_S seemed to fade fast since 3 h after IRI, an induction of p38 expression was not detectable (p-p38—IRI vs. IRI + 80 ppm H_2_S at 0 h: 1.21 ± 0.16, n.s., IRI vs. IRI + 80 ppm H_2_S at 3 h: 1.15 ± 0.17, n.s.). The slow-releasing H_2_S compound GYY 4137, administered intravenously immediately after IRI, significantly increased p38 phosphorylation (Figure 2b, p-p38—IRI: 1.09 ± 0.06 vs. IRI + GYY 4137: 1.33 ± 0.06, *p* < 0.01). This effect seemed more pronounced regarding ERK1/2 phosphorylation. Inhalation therapy with 80 ppm H_2_S showed a significant increase not only immediately after the treatment but also after a time delay of 1.5 and 3 h (Figure 3a, p-ERK1/2—IRI: 1.21 ± 0.10 vs. IRI + 80 ppm H_2_S at 0 h: 1.52 ± 0.08, *p* < 0.01, IRI vs. IRI + 80 ppm H_2_S at 1.5 h: 2.15 ± 0.21, *p* < 0.001, IRI vs. IRI + 80 ppm H_2_S at 3 h: 1.42 ± 0.19, *p* < 0.05). Treatment with GYY 4137 immediately following IRI was also associated with a significant increase in ERK-1/2 phosphorylation (Figure 3b, p-ERK1/2—IRI: 1.13 ± 0.06 vs. IRI + GYY 4137: 1.67 ± 0.09, *p* < 0.001). However, H_2_S post-conditioning had no effect on MAPK JNK (Data not shown).

Immunohistochemical staining of retinal cross-sections revealed the effect of IRI and H_2_S post-conditioning on ERK1/2 phosphorylation. After ischemia, only a discrete staining (Figure 4, green color) was detectable, whereas the following inhalative application of H_2_S (1.5 h) resulted in an increased expression in the area of the inner nuclear layer. In the controls this expression could not be observed. As an effect of IRI, up-regulation of GFAP was seen in ischemic eyes. However, we interpreted the strong GFAP signal after IRI and H_2_S as a transient Müller cell activation to protect retinal neurons and as a positive effect regarding neuronal regeneration.

### 2.3. H_2_S Prevents Apoptosis of Retinal Ganglion Cells after IRI

Effector caspase-3 also plays a crucial role in programmed cell death or apoptosis. We examined whether H_2_S post-conditioning protects RGCs against apoptosis by analyzing retinal caspase-3 cleavage after IRI and either 80 ppm H_2_S inhalation or intravenous application of GYY 4137. The 80 ppm H_2_S inhalation significantly reduced caspase-3 cleavage when initiated at 0 and 1.5 h after IRI (Figure 5a, Col 2, 3, IRI: 1.42 ± 0.13 vs. IRI + 80 ppm H_2_S at 0 h: 0.85 ± 0.09, *p* < 0.001, IRI vs. IRI + 80 ppm H_2_S at 1.5 h: 0.83 ± 0.15, *p* < 0.001), but not at 3 h after IRI (Figure 5a, Col 4, IRI vs. IRI + 80 ppm H_2_S at 3 h: 1.31 ± 0.13, n.s.). Intravenous application of GYY 4137 immediately after IRI resulted in significantly decreased caspase-3 cleavage (Figure 5b, IRI: 1.33 ± 0.10 vs. IRI + GYY 4137: 0.95 ± 0.11, *p* < 0.01).

### 2.4. H_2_S Suppresses IRI-Induced Increase in VEGF Expression

Unilateral IRI resulted in increased expression of the pro-angiogenetic growth factor VEGF. The 80 ppm H_2_S inhalation at 0 and 1.5 h, but not at 3 h after IRI, attenuated this increase (Figure 6a, IRI: 1.22 ± 0.13 vs. IRI + 80 ppm H_2_S at 0 h: 0.90 ± 0.14, *p* < 0.001, IRI vs. IRI + 80 ppm H_2_S at 1.5 h: 0.75 ± 0.13, *p* < 0.001, IRI vs. IRI + 80 ppm H_2_S at 3 h: 1.18 ± 0.11, n.s.). GYY 4137, administered intravenously immediately after IRI, also attenuated the increase in IRI-induced VEGF expression (Figure 6b, IRI: 1.22 ± 0.09 vs. IRI + GYY 4137: 0.84 ± 0.08, *p* < 0.01).

### 2.5. H2S Attenuates the Increase in Expression of Pro-Inflammatory Molecules ICAM-1 and iNOS following IRI

We assessed the influence of H_2_S on pro-inflammatory pathways after IRI by examining the expression of ICAM-1, a cell adhesion molecule that constitutes an integral part of leukocyte diapedesis during inflammatory processes, and the expression of iNOS, which is induced by pro-inflammatory cytokines. IRI led to an increased expression of ICAM-1 (Figure 7a, ICAM-1—IRI: 1.10 ± 0.09) and iNOS (Figure 8a, iNOS—IRI: 1.18 ± 0.04). The 80 ppm H_2_S inhalation was able to counteract this increase at 0 and 1.5 h, but not at 3 h after IRI (Figure 7a Col 2-4, ICAM-1—IRI vs. IRI + 80 ppm H_2_S at 0 h: 0.90 ± 0.10, *p* < 0.05, IRI vs. IRI + 80 ppm H_2_S at 1.5 h: 0.73 ± 0.09, *p* < 0.001, IRI vs. IRI + 80 ppm H_2_S at 3 h: 1.05 ± 0.19, n.s.; Figure 8a Col. 2-4, iNOS—IRI vs. IRI + 80 ppm H_2_S at 0 h: 1.02 ± 0.12, *p* < 0.01, IRI vs. IRI + 80 ppm H_2_S at 1.5 h: 0.92 ± 0.11, *p* < 0.01, IRI vs. IRI + H_2_S 80 ppm at 3 h: 1.22 ± 0.13, n.s.). The intravenous administration of GYY 4137 following IRI also attenuated the IRI-induced increase in ICAM-1 (Figure 7b, ICAM-1—IRI: 1.16 ± 0.06 vs. IRI + GYY 4137: 0.98 ± 0.08, *p* < 0.05) and iNOS expression (Figure 8b, iNOS—IRI: 1.17 ± 0.08 vs. IRI + GYY 4137: 0.93 ± 0.09, *p* < 0.05).

### 2.6. The Protective Effect of H_2_S Is Suppressed by the MEK/ERK Pathway Inhibitor PD 98059

As described previously, in retinal cross-sections after IRI and post-conditioning with inhalative H_2_S, ERK1/2 was detected predominately in the inner nuclear layer. This effect was abolished by the application of the MEK/ERK pathway inhibitor PD 98059 (Figure 9, green color). In the next step, we analyzed the RGC count after PD 98059. Our aim here was to test the hypothesis that the ERK 1/2 pathway, in particular, is involved in the protective effects of H_2_S. For these experiments, we used inhalative H_2_S because of its more potent effect. This protective effect was attenuated by ERK 1/2 inhibition prior to IRI (Figure 10 Col. 3, 4, IRI + 80 ppm H_2_S at 1.5 h: 2253 ± 177 vs. IRI + 80 ppm H_2_S at 1.5 h + PD 98059: 1722 ± 129, *p* < 0.001) suggesting a causal role of ERK 1/2 in H_2_S-mediated neuroprotection. Representative flat-mounts of FG-labeled RGC are shown in Figure 10.

### 2.7. Treatment with the MEK/ERK Pathway Inhibitor PD 98059 after IRI Abolishes H2S-Mediated Effects on the Heat Shock Response and Antioxidant Enzymes

Inhalative post-conditioning with 80 ppm H_2_S at 1.5 h after IRI significantly reduced the expression of HO-1 (Figure 11a, HO-1—IRI: 4.96 ± 0.66 vs. IRI + 80 ppm H_2_S at 1.5 h: 1.19 ± 0.21, *p* < 0.001), HSP-70 (Figure 11b, HSP-70—IRI: 1.79 ± 0.22 vs. IRI + 80 ppm H_2_S at 1.5 h: 1.17 ± 0.19, *p* < 0.001), SOD-1 (Figure 11c, SOD-1—IRI: 1.82 ± 0.22 vs. IRI + 80 ppm H_2_S at 1.5 h: 1.09 ± 0.11, *p* < 0.001), SOD-2 (Figure 11d, SOD-2—IRI: 1.57 ± 0.14 vs. IRI + 80 ppm H_2_S at 1.5 h: 1.22 ± 0.10, *p* < 0.001), and catalase (Figure 11e, catalase—IRI: 1.52 ± 0.13 vs. IRI + 80 ppm H_2_S at 1.5 h: 1.03 ± 0.14, *p* < 0.001) in the retina. These effects were significantly attenuated by the intravenous application of the MEK/ERK pathway inhibitor PD 98059 prior to IRI (Figure 11a, HO-1—IRI + 80 ppm H_2_S at 1.5 h: 1.19 ± 0.21 vs. IRI + 80 ppm H_2_S at 1.5 h + PD 98059: 4.45 ± 0.48, *p* < 0.001; Figure 11b, HSP-70—IRI + 80 ppm H_2_S at 1.5 h: 1.17 ± 0.19 vs. IRI + 80 ppm H_2_S at 1.5 h + PD 98059: 1.74 ± 0.14, *p* < 0.001; Figure 11c, SOD-1—IRI + 80 ppm H_2_S at 1.5 h: 1.09 ± 0.11 vs. IRI + 80 ppm H_2_S at 1.5 h + PD 98059: 1.72 ± 0.15, *p* < 0.001; Figure 11d, SOD-2—IRI + 80 ppm H_2_S at 1.5 h: 1.22 ± 0.10 vs. IRI + 80 ppm H_2_S at 1.5 h + PD 98059: 1.47 ± 0.08, *p* < 0.01; Figure 11e, Catalase—IRI + 80 ppm H_2_S at 1.5 h: 1.03 ± 0.14 vs. IRI + 80 ppm H_2_S at 1.5 h + PD 98059: 1.29 ± 0.14, *p* < 0.01).

## 3. Discussion

The aim of this study was to identify molecular pathways underlying the neuroprotective effects of H_2_S applied at different times and by different modes of application after retinal IRI. The main findings can be summarized as follows: (1) post-conditioning with 80 ppm H_2_S and intravenous GYY 4137 increased the phosphorylation of the MAP kinases p38 and ERK-1/2; (2) post-conditioning with 80 ppm H_2_S and intravenous GYY 4137 decreased caspase-3 cleavage to promote an anti-apoptotic effect; (3) post-conditioning with 80 ppm H_2_S and intravenous GYY 4137 reduced the expression of ICAM-1 and iNOS, which are key components of pro-inflammatory signaling cascades; (4) post-conditioning with H_2_S supported retinal ganglion cell survival after IRI, with inhalation appearing superior to the intravenous application; (5) conversely, treatment with the ERK pathway inhibitor PD 98059 suppressed the protective effect of H_2_S, strongly suggesting that at least part of the neuroprotective effect of H_2_S is mediated through the ERK pathway; (6) post-conditioning with 80 ppm H_2_S inhibited the heat shock response and formation of free radical scavengers, and this effect was abrogated by PD 98059.

The phosphorylation of mitogen-activated protein kinases (p38, ERK-1/2, and JNK) plays an integral role in many intracellular pathways including apoptosis, inflammation, and hypoxia response [19]. The MAPK pathway has been implicated in neuroprotection as a potential therapeutic target in the context of ischemic and hemorrhagic cerebral vascular disease [20]. However, the available data concerning the effect of H_2_S on MAPK signaling are contradictory, most likely because of the use of artificial cell experiments in different strains and diverse H_2_S concentrations [21]. In the context of IRI, H_2_S has been shown to promote the phosphorylation of ERK-1/2 after coronary artery occlusion and reperfusion in pigs [22] and to reduce intestinal leukocyte attachment and rolling in postcapillary venules by p38 activation in a mouse model of IRI [23]. Biermann et al. showed that pre-conditioning with inhaled H_2_S increased the phosphorylation of JNK, but decreased the phosphorylation of ERK-1/2 after retinal IRI, whereas p38 phosphorylation was not affected [17]. Conversely, post-conditioning with the fast H_2_S-releasing compound NaHS protected hippocampal neurons against injury in a rat model of transient global cerebral ischemia and this effect was at least partially mediated via ERK-1/2 activation [24]. Our study findings in rats substantiate these results, as we observed increased phosphorylation of ERK-1/2 and p38 after retinal IRI and post-conditioning with H_2_S.

Part of the protective effect of H_2_S seems attributable to an inhibition of apoptosis. In several animal disease models, H_2_S decreased the expression of pro-apoptotic caspases, increasing the expression of anti-apoptotic mediators, such as Bcl-2. Caspases, which are synthesized as inactive proenzymes, play essential roles in apoptosis, and are regulated by the pro- and anti-apoptotic proteins of the Bcl-2 family and heat shock proteins [25]. Liu et al. showed that treatment with H_2_S in the form of NaHS decreased the expression of caspase-3, increased the expression of Bcl-2, and attenuated myocardial fibrosis in diabetic rats [26]. Exposure of rats to NaHS decreased caspase-3 levels and protected hippocampal neurons against lead-mediated neuronal damage [27]. Biermann et al. reported that rapid pre-conditioning with inhaled H_2_S significantly reduced caspase-3 cleavage after retinal IRI [17]. In the current investigation, we also detected a significant decrease in retinal caspase-3 cleavage after post-conditioning with inhaled H_2_S at a concentration of 80 ppm when initiated immediately after and at 1.5 h after IRI, as well as after treatment with GYY 4137, thereby confirming the anti-apoptotic effect of H_2_S.

Since ischemia-reperfusion injury induced the upregulation of iNOS via MAPKs, excessive release of oxygen radicals caused oxidative stress in the ischemic area [28,29]. In the retina, upregulation of iNOS by IRI was associated with apoptosis, cell migration, and endothelial cell dysfunction [30]. Furthermore, increased iNOS expression triggered the synthesis of the inflammatory mediator ICAM-1 [31]. The IRI-induced inflammatory response initiates the upregulation of VEGF, a potent vascular permeability regulator associated with vasogenic and cytotoxic edema in the acute phase of ischemic injury and enhanced cell death [32].

H_2_S has been described as a novel mediator of inflammation in various disease models [33,34,35,36] and appears to affect iNOS, ICAM-1, and VEGF. Experiments using a rat model of streptozotocin-induced diabetic retinopathy showed that H_2_S treatment attenuated the increased expression of IL-1ß, ICAM-1, iNOS, and COX-2 [14,37]. In the present investigation, the expression of ICAM-1 and iNOS was increased after retinal IRI, whereas post-conditioning with H_2_S attenuated the IRI-induced increase in ICAM-1 and iNOS expression. We also investigated the impact of H_2_S post-conditioning on the retinal expression of VEGF after IRI. IRI elicits an inflammatory response that impairs the endothelial barrier function and increases microvascular permeability [38]. VEGF is a major contributor to angiogenesis and vascular permeability; therefore, its secretion increases in response to an ischemic insult [39,40]. H_2_S treatment of rats with streptozotocin-induced diabetic retinopathy reduced the vitreal VEGF content [14]. We observed a similar suppression of retinal VEGF expression after IRI and H_2_S treatment. In summary, our study confirmed the protective effects of H_2_S reducing iNOS, ICAM-1, and VEGF.

In a previous study we demonstrated that post-conditioning with H_2_S inhalation shows time- and dose-dependent effectiveness in protecting RGCs from cell death after IRI [18]. We found that the most striking results were achieved with the application of 80 ppm H_2_S at 0 or 1.5 h after retinal IRI. We therefore chose 80 ppm H_2_S as the dosage for all further experiments involving inhalative H_2_S application. Our comparison of the effects of inhalative post-conditioning and intravenous treatment with the H_2_S donor GYY 4137 revealed neuroprotective effects with both methods of application. However, the results were more pronounced after H_2_S inhalation. The inhibition of ERK-1/2 using PD98059 attenuated the protective effects of H_2_S on RGC survival after retinal IRI, and PD 98059 abolished the H_2_S-related reduction of the heat shock response and of reactive oxygen species (ROS) scavenger systems, indicating increased cell stress. Therefore, we conclude that at least part of the neuroprotective effect of H_2_S post-conditioning on retinal IRI is mediated via the ERK pathway.

The heat shock response, originally described to occur in response to heat (hence the name), is regarded as a cellular protective mechanism, but it has also been linked to a variety of other stressors, including IRI [20,41]. Harmful events activate the heat shock response which then protects cells from further harm. Part of this cytoprotective effect may be related to the ability of heat shock proteins (HSPs) to stabilize intracellular protein structures [38,42], as HSP-70 induction has been implicated in neuroprotection after brain injury [43]. HO-1, a marker of cell stress, is involved in cellular defense mechanisms and aids in preventing further cell damage [44].

Pre-conditioning with H_2_S was associated with induction of the heat shock response and increased expression of HSP-90 in a rat model of retinal IRI [17]. In the present study, the effect of H_2_S treatment on this heat shock response after retinal IRI was investigated. We observed a marked increase in the expression of HSP-70 and HO-1 after IRI, but this expression was significantly attenuated by 80 ppm H_2_S inhalation at 1.5 h after IRI. We conclude from these findings that H_2_S post-conditioning can prevent cell stress after IRI. Inhibition of the ERK pathway abolished this protective effect, suggesting a key role for this signaling pathway in the protective effect of H_2_S.

Free radical damage is known to play a critical role in cerebral ischemia-reperfusion injury, and antioxidant enzymes, such as superoxide dismutase and catalase, can protect tissues against the cytotoxicity caused by ROS [19,45]. ROS affect several major intracellular signaling pathways, including the MAPK pathway [19]. In the present study, IRI significantly increased the levels of the ROS-scavenging enzymes SOD-1, SOD-2, and catalase 24 h after IRI. This finding was in agreement with the results presented by Lewden et al. who described a significant increase in catalase activity 15 and 120 min after ischemia and reperfusion of the rat retina [46]. Conversely, Li et al. found diminished levels of SOD, catalase, and glutathione peroxidase 7 days after retinal IRI [47]. These findings seem to suggest a differential effect of retinal IRI on the expression of ROS-scavenging systems depending on the time elapsed since IRI. In the present study, treatment with H_2_S significantly reduced the activities of SOD-1, SOD-2, and catalase. We conclude from this that H_2_S treatment prevents oxidative stress, thereby reducing the requirement/demand for free radical scavenging systems and suppressing their activity.

H_2_S has been described as a potent antioxidant in several organ systems [48,49,50]. In a murine model of acute lung injury induced by LPS-induced endotoxemia, the H_2_S donor GYY 4137 attenuated the LPS-induced decrease in the antioxidative biomarkers catalase and SOD and the total antioxidant capacity (T-AOC) [51]. Kimura et al. showed that H_2_S promotes the production of glutathione, an important antioxidant and part of the cellular defense system against oxidative stress in the CNS [52]. Interestingly, we saw a significant reduction in the mRNA activities of SOD-1, SOD-2, and catalase after retinal IRI and treatment with H_2_S. This effect was attenuated by additional treatment with PD 98059, suggesting a causal role of the MAP kinase pathway in facilitating the effects of H_2_S on the oxidative stress response. A potential explanation for this finding could be that retinal IRI rapidly activates antioxidant enzymes as a counter-regulation in response to increased demand. We view the reduced levels of ROS-scavenging enzymes after treatment with H_2_S as a sign of reduced oxidative stress, thereby highlighting the protective effect of H_2_S. However, the role of H_2_S in IRI and oxidative stress needs to be evaluated more extensively.

This study has some limitations to consider. In the present study, different time points were chosen for H_2_S administration with the best effect observed at a time delay of 1.5 h after the end of ischemia, whereas intravenous GYY4137 was administered only just before the end of ischemia. Although GYY4137 is a molecule that releases H_2_S slowly, the effect of a later application remains unclear. In addition, we administered only one concentration of GYY4137 (25 mg/kg i.v.) and a higher dosage might be beneficial.

## 4. Materials and Methods

### 4.1. Animals

Adult male and female Sprague-Dawley rats (male to female ratio 1:1, body weight 280–350 g, Charles River, Sulzfeld, Germany) were used for the experiments. All animals were fed a standard rodent diet ad libitum and kept on a 12 light/12 h dark cycle. Approval for the experiments was obtained a priori from the Committee for Animal Care of the University of Freiburg (approval number: 35-9185.81/G-16/46). All interventions were conducted in accordance with the ARRIVE guidelines for the use of animals in research.

All operations and manipulations were performed under general anesthesia, as previously described by Goebel et al. [53,54]. The number of animals used for RGC quantification and molecular analyses was *n* = 8 or *n* = 6 per group, respectively.

### 4.2. Retrograde Labeling of RGC

After induction of anesthesia by isoflurane inhalation, animals were placed in a stereotactic apparatus (Stoelting, Kiel, Germany). The skin covering the skull was cut open and retracted, and the Lambda and Bregma sutures were used as landmarks to drill three holes into the parietal bone on either side of the sagittal suture. A total amount of 7.8 µL fluorogold (FG), dissolved in dimethyl sulfoxide/phosphate buffered saline (PBS), was then injected through the drill holes into both superior colliculi. Animals received subcutaneous buprenorphine for postoperative pain control. Adequate retrograde transport of FG into the perikarya of the RGCs was ensured by waiting for 7 days after RGC labeling to conduct further experiments.

### 4.3. Retinal Ischemia-Reperfusion Injury and H_2_S Treatment

Animals were anesthetized by intraperitoneal application of a mixture of xylazine and ketamine. The anterior chamber of the left eye was then cannulated with a 30G needle connected to a reservoir containing 0.9% NaCl. The hydrostatic pressure of the water column was used to raise the intraocular pressure to 120 mmHg. Retinal ischemia was verified by microscopy inspection of the retina to confirm interrupted retinal blood flow. Retinal ischemia was maintained for 60 min and was then promptly terminated by removal of the needle tip. Immediate restoration of retinal perfusion again was ascertained by microscopy. Animals with insufficient recovery of retinal perfusion were excluded from further analysis, as were those with lens injury since lens injury prevents RGC death and promotes axonal regeneration [55].

The rats were then randomized into 4 groups: group 1 received room air, group 2 received H_2_S at 80 ppm in 21% oxygen for 60 min at 1.5 h after ischemia and reperfusion, group 3 received the slow-releasing H_2_S donor GYY4137 (Abcam, Cambridge, UK) (25 mg/kg) intravenously, and group 4 received the selective MEK/ERK pathway inhibitor PD98059 intravenously 60 min prior to induction of retinal ischemia followed by H_2_S application, as described for group 2. GYY4137 or PD98059 were applied by establishing intravenous access via cannulation of the tail vein with a 24G needle.

### 4.4. RGC Quantification

Animals were euthanized by CO_2_ inhalation 7 days after IRI. For whole-mount preparation of the retinas, retinal tissue was immediately removed and placed in ice-cold Hank’s balanced salt solution prior to further processing. Retinas were then carefully placed on a nitrocellulose membrane with the ganglion cell layer (GCL) facing upward. Remnants of the vitreous body were removed and the retinas were fixed in 4% paraformaldehyde for 1 h and then embedded in mounting medium (Vectashield; Axxora, Loerrach, Germany).

The densities of FG-positive RGCs were determined by fluorescence microscopy (AxioImager; Carl Zeiss, Jena, Germany) and the appropriate bandpass emission filter (FG: excitation/emission, 331/418 nm). The examination of retinas was performed in a blinded fashion, as previously described [54,56]. In brief, we photographed 3 standard rectangular areas (0.2 mm × 0.2 mm = 0.04 mm^2^) in the central regions of each retinal quadrant at a distance of 1, 2, and 3 mm from the optic disc. Consequently, an area of 0.48 mm^2^ per retina was evaluated. The average RGC density in cells/mm^2^ was calculated by multiplying the number of cells analyzed/0.04 mm^2^ by 25. Secondary FG staining can occur in activated microglial cells after RGC phagocytosis; therefore, those cells were identified based on morphologic criteria and excluded from the analysis.

### 4.5. Real-Time Polymerase Chain Reaction

The rat eyes were enucleated 24 h after IRI for harvest of retinal tissue. Total RNA from ¼ of the retina was extracted using a column-based purification kit (RNeasy Micro Kit, Qiagen, Hilden, Germany), according to the manufacturer’s instructions. A 50 ng sample of total RNA was utilized for reverse transcription using random primers (High capacity cDNA Reverse Transcription Kit, Thermo Fisher Scientific, Waltham, MA, USA). RT-PCR was performed with a TaqMan^®^ probe-based detection kit (TaqMan^®^ PCR Universal Master Mix, Thermo Fisher Scientific, Waltham, MA, USA) and the following primers: HO-1 Rn00561387_m1, HSP-70 Rn04224718_u1, SOD-1 Rn00566938_m1, SOD-2 Rn00690588_g1, catalase Rn01512559_m1, and GAPDH Rn01775763_g1 (all from Thermo Fisher Scientific, Waltham, MA, USA).

We employed an RT-PCR system (StepOnePlus^®^, Thermo Fisher Scientific, Waltham, MA, USA) with the following cycling conditions to conduct the PCR assays: 95 °C for 10 min, 40 cycles of 95 °C for 10 s, and 60 °C for 1 min. The reaction specificity was validated by running appropriate negative controls. The results were interpreted by normalizing the cycle of threshold (CT) values for each gene of interest to the corresponding CT values for glyceraldehyde 3-phosphate dehydrogenase (GAPDH) (∆CT). The relative gene expression (∆∆CT) in IR injured retinal tissue in the four treatment groups was calculated in relation to the corresponding gene expression in untreated retinas of each individual animal.

### 4.6. Immunhistochemical Staining

At 24 h after IRI, the rat eyes were enucleated and immediately placed in 4% paraformaldehyde for 1 h at 4 °C. The eyes were then washed in Dulbecco’s phosphate buffered saline (D-PBS), post-fixed in 20% sucrose for 4 h at room temperature, and washed again in D-PBS. The eyes were then embedded in mounting medium (Tissue-Tek^®^ O.C.T. Compound, Sakura-Finetek, Torrance, CA, USA) and frozen in liquid nitrogen.

For immunohistochemistry, histological sections (10 µm) were cut through the middle third of the eyes using a cryostatic temperature regulator and collected on gelatinized microscope slides. Immunohistochemistry was performed according to standardized protocols, as previously described by Biermann et al. [17]. Immunolabeling was performed with primary antibodies against GFAP (#sc-33673, Santa Cruz Biotechnology, Dallas, TX, USA) and p-ERK1/2 (#9101, Cell Signaling, Technology, Danvers, MA, USA). Primary antibodies were then conjugated with their corresponding secondary antibodies (red fluorescence: #115-166-062, Cyanine Cy3, goat anti-mouse, dilution 1:500; green fluorescence: #111-545-144, Alexa Fluor 488, goat anti-rabbit; both Jackson ImmunoResearch Laboratories Inc., West Grove, PA, USA). Nuclei were counterstained with 4′,6-diamino-2-phyenylindole dihydrochloride hydrate (DAPI; Sigma Aldrich, St. Louis, MO, USA).

The slides were examined, and relevant sections were photographed using a fluorescence microscope (Axioplot, Carl Zeiss, Jena, Germany).

### 4.7. Western Blot Analysis

Retinal tissue was harvested 24 h after IRI to investigate protein expression. Total protein from 75% of the retina was extracted and processed for Western blotting, as previously described [56,57]. The membranes were blocked with 5% skim milk or bovine serum albumin in Tween20/PBS to minimize non-specific antibody binding and were subsequently incubated overnight with the following protein-specific antibodies, diluted as recommended by the manufacturer (iNOS#130246, VEGF#102643, ICAM-1#100450, caspase 3 #110543, cleaved caspase #86952, all Genetex, Irvine, CA, USA; ß-actin #3700, p-p38 #4511, p38 8690, p-ERK1/2 #9101, ERK1/2 #4695, all Cell Signaling Technology, Danvers, MA, USA). After incubation with a horseradish peroxidase-conjugated anti-rabbit secondary antibody (GE Healthcare, Freiburg, Germany), the proteins were visualized with an enhanced chemiluminescence Western blotting detection reagent (Western Lightning plus enhanced chemiluminescence, #NEL104001EA, PerkinElmer, Waltham, MA, USA), according to the manufacturer’s instructions to visualize the proteins. Relative changes in protein expression in retinas exposed to IRI in the four treatment groups were calculated based on the untreated retinas of each respective animal. The chemiluminescence imaging system Fusion Fx^®^ (Vilber, Collegién, France) was employed for recordings and densitometric analysis.

### 4.8. Statistical Analysis

Data were evaluated using a computerized statistics program (SigmaPlot Version 11.0, Systat Software Inc., San Jose, CA, USA). All data are presented as mean values ± standard deviation (SD). One-way ANOVA was employed for between-group comparisons using the post hoc Holm-Sidak test and Kruskal–Wallis test for data that lacked normal distribution. A *p* value < 0.05 was considered statistically significant.

## 5. Conclusions

In the present work, we demonstrated that the neuroprotective effect of either inhaled H_2_S or H_2_S released by GYY 4137 is predominantly mediated by the ERK1/2 pathway. However, H2S also affects other molecules involved in apoptotic and inflammatory processes. When compared with the GYY 4137 treatment, inhaled H_2_S seems to achieve a stronger neuroprotective effect. Inhaled H_2_S reduces the heat shock response and the expression of radical scavengers. Further research is needed to fully elucidate the underlying mechanisms of action, but H_2_S appears to have potent neuroprotective action. Nevertheless, H_2_S could conceivably be used in the future for the clinical treatment of neuronal ischemia-reperfusion injury, such as glaucoma or stroke, to reduce neuronal cell death. Although potentially less effective, intravenous administration seems to be advantageous in practice due to concerns regarding the storage and handling of this potentially toxic gas.

## Figures and Tables

**Figure 1 ijms-23-05519-f001:**
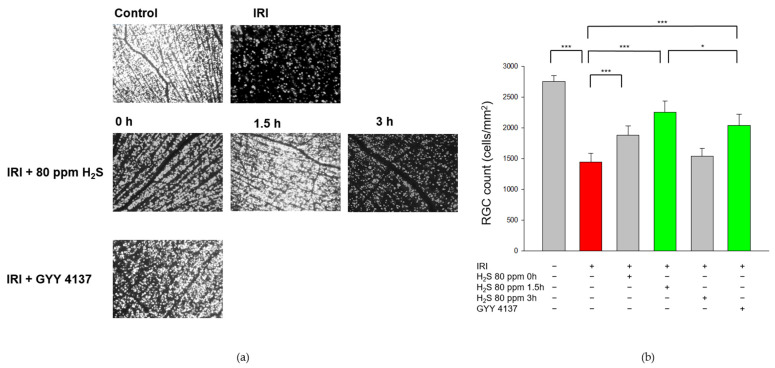
Effect of H_2_S on retinal ganglion cell count after ischemia-reperfusion injury (IRI). Rats were subjected to unilateral retinal IRI and subsequently either received inhalative therapy with 80 ppm H_2_S directly and 1.5 h after IRI or an intravenous application of the slow-releasing H_2_S donor GYY 4137 immediately following IRI. (**a**) Representative flat mound images of fluoroscope-labeled retinal ganglion cells 7 days after IRI and respective treatment. (**b**) Quantification of retinal ganglion cell density [cells/mm^2^, data are mean ± SD, *** = *p* < 0.001, untreated vs. IRI (*n* = 8), IRI vs. IRI + 80 ppm H_2_S at 0 h (*n* = 8), vs. IRI + 80 ppm H_2_S at 1.5 h (*n* = 8), and vs. IRI + GYY 4137 (*n* = 6)]. The total numbers of surviving ganglion cells after inhalation of H_2_S with a delay of 1.5 h was significantly increased compared to intravenous treatment with GYY 4137 [* = *p* < 0.05, IRI + 80 ppm H_2_S at 1.5 h (*n* = 8) vs. IRI + GYY 4137 (*n* = 6)]. The red column illustrates the reduction of RGC by IRI, and the green columns highlight the positive effect of H_2_S and GYY.

**Figure 2 ijms-23-05519-f002:**
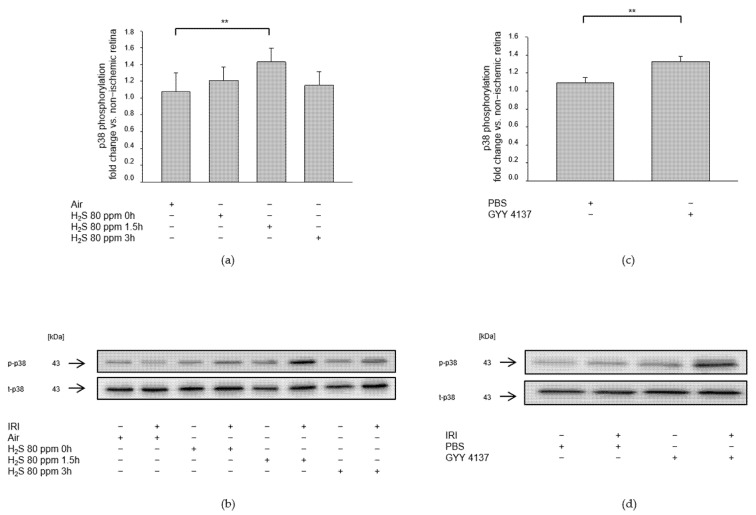
Effect of H_2_S on phosphorylation of retinal p38 after ischemia-reperfusion injury (IRI). Rats were treated either with inhalation of 80 ppm H_2_S for 60 min at various time points after IRI or with the intravenous application of GYY 4137 or the vehicle PBS (phosphate buffer saline) alone immediately following IRI. Retinal homogenates were used for Western blot analysis. (**a**,**c**) Densitometric analysis of p38 phosphorylation after inhalation of 80 ppm H_2_S at 0, 1.5, and 3 h after IRI (data are mean ± SD, *n* = 8, ** = *p* < 0.01, IRI vs. IRI + 80 ppm H_2_S at 1.5 h) and after IRI and intravenous application of GYY 4137 (data are mean ± SD, *n* = 6, IRI vs. IRI + GYY 4137). (**b**,**d**) Representative Western blot images illustrating the increment in p38 phosphorylation after IRI and H_2_S inhalation (*n* = 8) and after IRI and treatment with GYY 4137 (*n* = 6).

**Figure 3 ijms-23-05519-f003:**
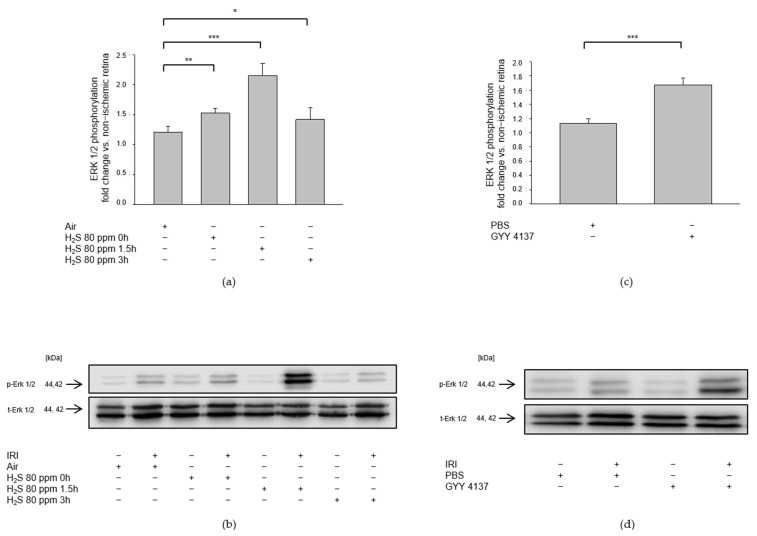
Effect of H_2_S on retinal ERK (extracellular signal-regulated kinase) 1/2 phosphorylation after ischemia-reperfusion injury (IRI). Rats were treated either with inhalation of 80 ppm H_2_S for 60 min at various time points after IRI or with the intravenous application of the slow-releasing H_2_S donor GYY 4137 immediately following IRI. Retinal homogenates were used for Western blot analysis. (**a,c**) Densitometric analysis of ERK1/2 phosphorylation after inhalation of 80 ppm H_2_S at 0, 1.5, and 3 h after IRI (data are mean ± SD, *n* = 8, ** = *p* < 0.01, IRI vs. IRI + 80 ppm H_2_S at 0 h, *** = *p* < 0.001, IRI vs. IRI + 80 ppm H_2_S at 1.5 h, and * = *p* < 0.05, IRI vs. IRI + 80 ppm H_2_S at 3 h after IRI) and after IRI and treatment with GYY 4137 (data are mean ± SD, *n* = 6, *** = *p* < 0.001, IRI vs. IRI + GYY 4137). (**b,d**) Representative Western blot images showing the enhancement of ERK 1/2 phosphorylation after IRI and H_2_S inhalation (*n* = 8) and after IRI and treatment with GYY 4137 (*n* = 6).

**Figure 4 ijms-23-05519-f004:**
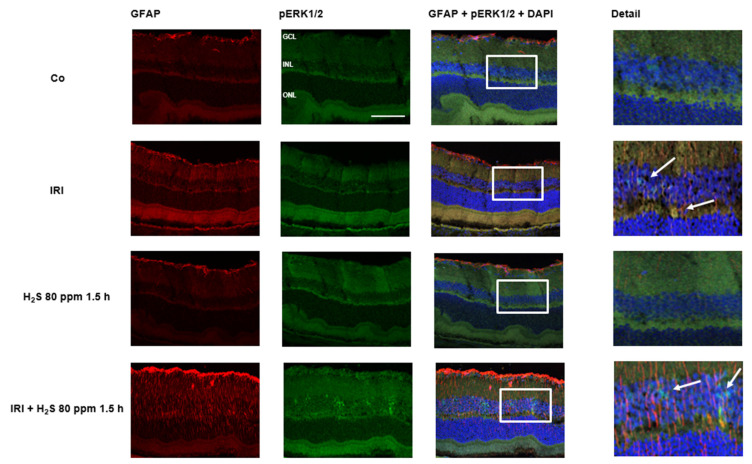
Retinal GFAP (glial fibrillary acidic protein) and pERK (phosphorylated extracellular signal-regulated kinase) 1/2 after unilateral ischemia-reperfusion injury (IRI) and H_2_S inhalation. Cell nuclei were counterstained with DAPI (4′,6-diamidino-2-phenylindole). Exposure to H_2_S led to an increase in phosphorylation of ERK1/2 after IRI, mainly in the inner nuclear layer (Scale bar, 50 µm; GCL, ganglion cell layer; INL, inner nuclear layer; ONL, outer nuclear layer). The white arrows indicate ERK1/2 phosphorylation in the INL.

**Figure 5 ijms-23-05519-f005:**
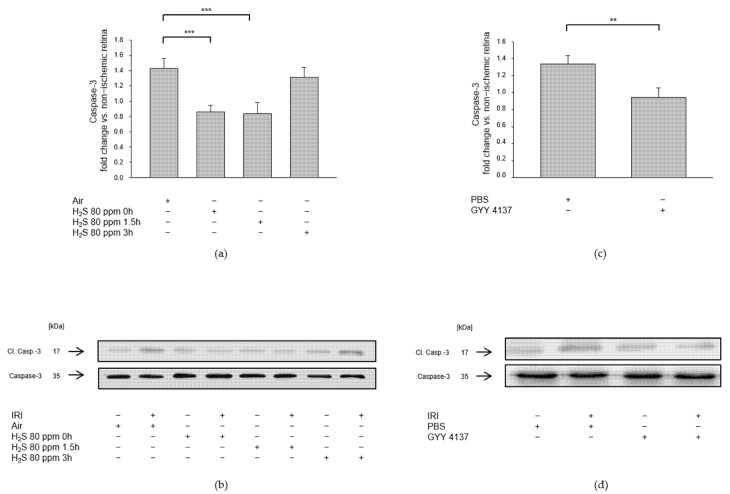
Effect of H_2_S on caspase-3 cleavage (cl. Casp.-3) after ischemia-reperfusion injury (IRI). Rats were treated either with inhalation of 80 ppm H_2_S for 60 min at various time points after IRI or by intravenous application of the slow-releasing H_2_S donor GYY 4137 immediately following IRI. Retinal homogenates were used for Western blot analysis. Densitometric analyses were normalized to total caspase-3 protein expression for all samples. (**a,c**) Densitometric analysis of retinal caspase-3 cleavage after inhalative post-conditioning with 80 ppm H_2_S at 0, 1.5, and 3 h after IRI (data are mean ± SD, *n* = 8, *** = *p* < 0.001, IRI vs. IRI + 80 ppm H_2_S at 0 h, and vs. IRI + 80 ppm H_2_S at 1.5 h after IRI) and after IRI and treatment with GYY 4137 (data are mean ± SD, *n* = 6, ** = *p* < 0.01, IRI vs. IRI + GYY 4137). (**b,d**) Representative Western blot images showing the decrease in caspase-3 cleavage after IRI and H_2_S inhalation (*n* = 8) and after IRI and treatment with GYY 4137 (*n* = 6).

**Figure 6 ijms-23-05519-f006:**
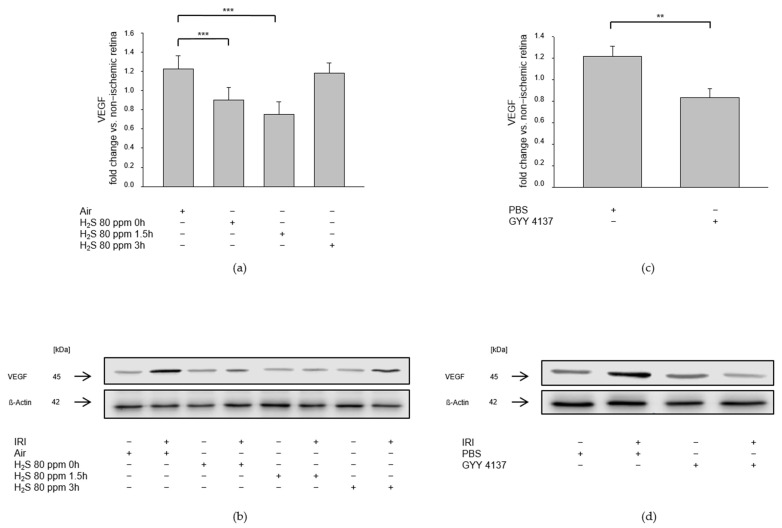
Effect of H_2_S on expression of VEGF (vascular endothelial growth factor) after ischemia-reperfusion injury (IRI). Rats were treated either with inhalation of 80 ppm H_2_S for 60 min at various time points after IRI or with the intravenous application of the slow-releasing H_2_S donor GYY 4137 immediately following IRI. Retinal homogenates were used for Western blot analysis. Densitometric analyses were normalized to ß-actin for all samples. (**a,c**) Densitometric analysis for the expression of VEGF after 80 ppm H_2_S inhalation at 0, 1.5, and 3 h after IRI (data are mean ± SD, *n* = 8, *** = *p* < 0.001, IRI vs. IRI + 80 ppm H_2_S at 0 h, and vs. IRI + 80 ppm H_2_S at 1.5 h after IRI) and after IRI and treatment with GYY 4137 (data are mean ± SD, *n* = 6, ** = *p* < 0.01, IRI vs. IRI + GYY 4137). (**b,d**) Representative Western blot images illustrating the attenuation of IRI-induced increase in VEGF expression after inhalation of H_2_S (*n* = 8) and after intravenous application of GYY 4137 (*n* = 6).

**Figure 7 ijms-23-05519-f007:**
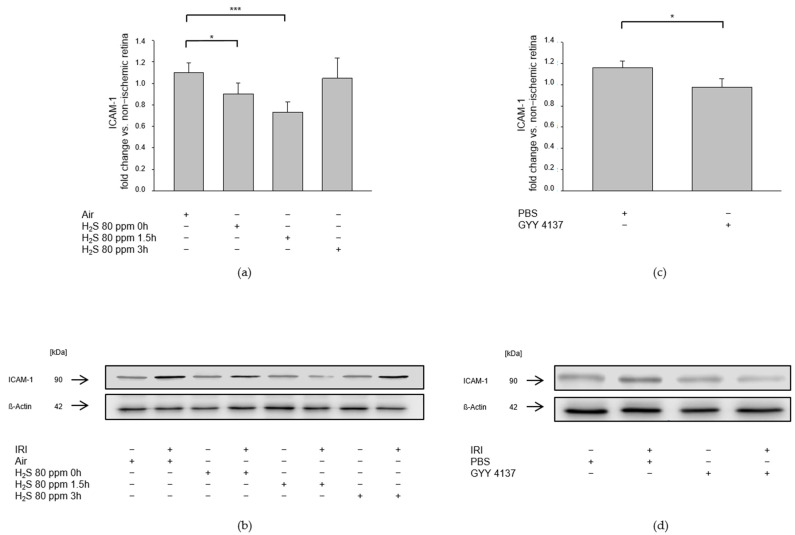
Effect of H_2_S on retinal ICAM-1 (intercellular adhesion molecule 1) expression after ischemia-reperfusion injury (IRI). Rats were treated either with inhalation of 80 ppm H_2_S for 60 min at various time points after IRI or with the intravenous application of the slow-releasing H_2_S donor GYY 4137 immediately following IRI. Retinal homogenates were used for Western blot analysis. Densitometric analyses were normalized to ß-actin for all samples. (**a,c**) Densitometric analysis for the expression of ICAM-1 after 80 ppm H_2_S inhalation at 0, 1.5, and 3 h after IRI (data are mean ± SD, *n* = 8, * = *p* < 0.05, IRI vs. IRI + 80 ppm H_2_S at 0 h, *** = *p* < 0.01, IRI vs. IRI + 80 ppm H_2_S at 1.5 h) and after IRI and treatment with GYY 4137 (data are mean ± SD, *n* = 6, * = *p* < 0.05, IRI vs. IRI + GYY 4137). (**b,d**) Representative Western blot images showing increased ICAM-1 expression after IRI which is attenuated by 80 ppm H_2_S-inhalation at 0 and 1.5 h after IRI (*n* = 8) and after application of the H_2_S donor GYY 4137 (*n* = 6).

**Figure 8 ijms-23-05519-f008:**
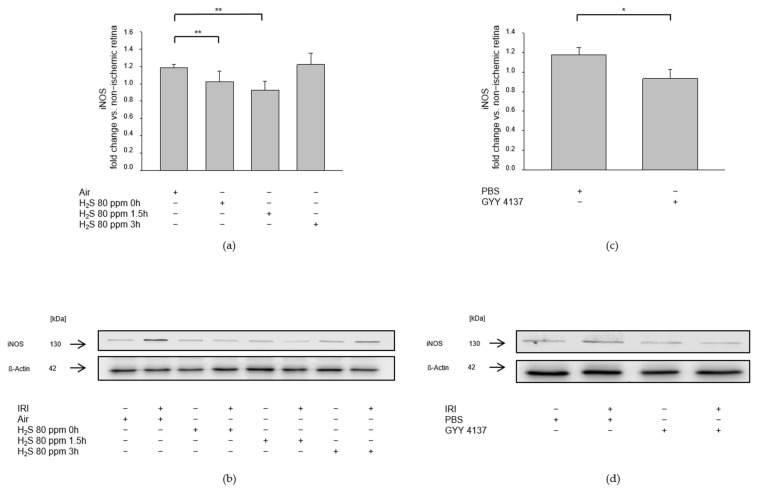
Effect of H_2_S on expression of iNOS (inducible nitric oxide synthase) after ischemia-reperfusion injury (IRI). Rats were treated either with inhalation of 80 ppm H_2_S for 60 min at various time points after IRI or with the intravenous application of the slow-releasing H_2_S donor GYY 4137 immediately following IRI. Retinal homogenates were used for Western blot analysis. Densitometric analyses were normalized to ß-actin for all samples. (**a**,**c**) Densitometric analysis for the expression of iNOS after 80 ppm H_2_S inhalation at 0, 1.5, and 3 h after IRI (data are mean ± SD, *n* = 8, ** = *p* < 0.01, IRI vs. IRI + 80 ppm H_2_S at 0 h, and vs. IRI + 80 ppm H_2_S at 1.5 h) and after IRI and treatment with GYY 4137 (data are mean ± SD, *n* = 6, * = *p* < 0.05, IRI vs. IRI + GYY 4137). (**b**,**d**) Representative Western blot images illustrating the induction of iNOS expression by IRI which is attenuated by 80 ppm H_2_S inhalation at 0 and 1.5 h after IRI (*n*= 8) and by treatment with GYY 4137 (*n* = 6).

**Figure 9 ijms-23-05519-f009:**
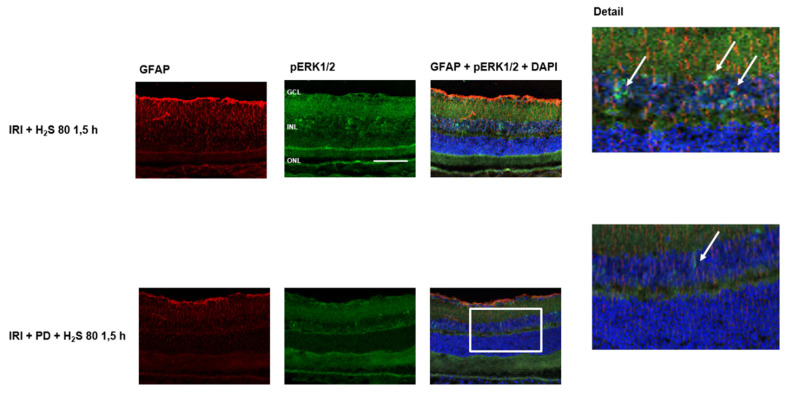
Retinal GFAP (glial fibrillary acidic protein) and pERK 1/2 (phosphorylated extracellular signal-regulated kinase) after unilateral ischemia-reperfusion injury (IRI) and H_2_S inhalation with application of the MEK/ERK pathway inhibitor PD 98059. Cell nuclei were counterstained with DAPI. Exposure to H_2_S led to an increase in phosphorylation of ERK1/2 after IRI, mainly in the inner nuclear layer. This effect was almost completely attenuated by the MEK/ERK pathway inhibitor PD 98059 (Scale bar, 50 µm; GCL, ganglion cell layer; INL, inner nuclear layer; ONL, outer nuclear layer). The white arrows indicate ERK1/2 phosphorylation in the INL.

**Figure 10 ijms-23-05519-f010:**
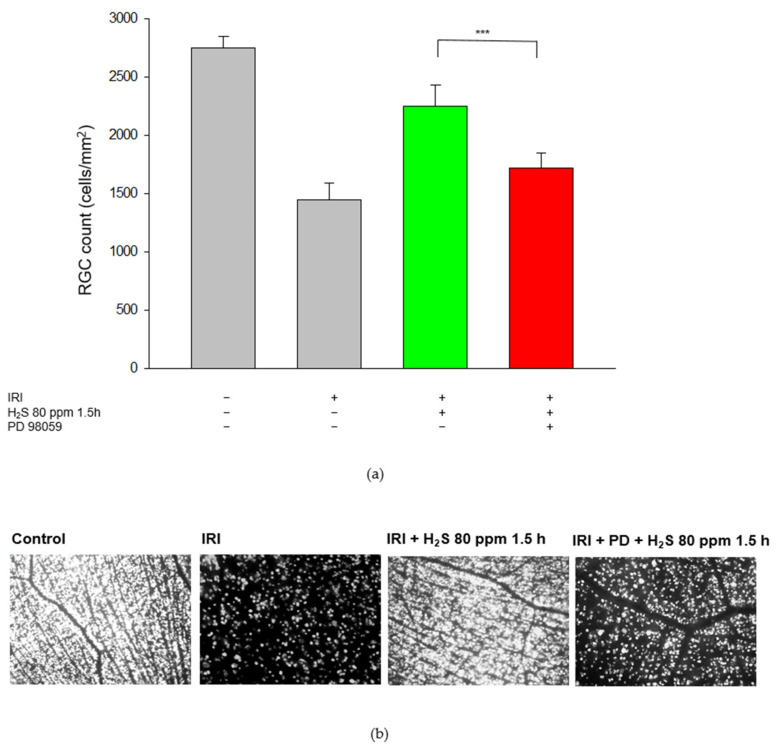
Effect of H_2_S and the ERK (extracellular signal-regulated) pathway inhibitor PD 98059 on retinal ganglion cells after ischemia-reperfusion injury (IRI) and inhalative therapy with 80 ppm H_2_S at 1.5 h after IRI, with or without the addition of ERK pathway inhibitor PD 98059. (**a**) Representative flat mound images of fluoroscope-labeled retinal ganglion cells 7 days after IRI and respective treatment. (**b**) Quantification of retinal ganglion cell density (cells/mm^2^, data are mean ± SD, *** = *p* < 0.001, IRI + 80 ppm H_2_S vs. IRI + 80 ppm H_2_S + PD 98059 (*n* = 6)). The green column illustrates the positive effect of H_2_S on IRI and the red column highlights the inhibition of PD98059.

**Figure 11 ijms-23-05519-f011:**
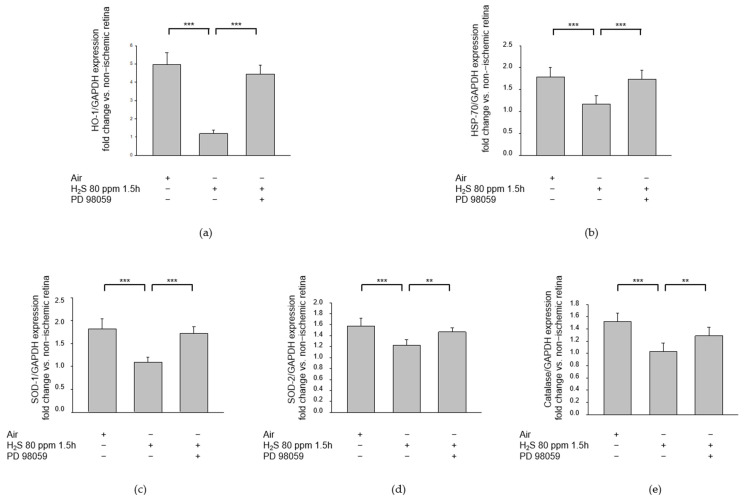
Effect of H_2_S and the ERK (extracellular signal-regulated) pathway inhibitor PD 98059 on retinal heat shock response after ischemia-reperfusion injury (IRI). Rats were subjected to unilateral retinal ischemia-reperfusion injury (IRI) and subsequently received inhalative therapy with 80 ppm H_2_S at 1.5 h after IRI, with or without the intravenous administration of ERK pathway inhibitor PD 98059. Retinal homogenates were used for analysis by quantitative real-time PCR. All results were normalized to GAPDH. (**a**–**e**) Fold induction of (**a**) HO-1 (heme oxygenase 1) mRNA expression, (**b**) HSP-70 (heat shock protein 70) mRNA expression, (**c**) SOD-1 (superoxide dismutase 1) mRNA expression, (**d**) SOD-2 (superoxide dismutase 2) mRNA expression, and (**e**) catalase mRNA expression after IRI. An 80 ppm H_2_S inhalation at 1.5 h after IRI was able to attenuate this increase in heat shock response. This effect was counteracted by the ERK pathway inhibitor PD 98059. (Data are mean ± SD, *n* = 8 for IRI and IRI + 80 ppm H_2_S, *n* = 6 for IRI + 80 ppm H_2_S + PD 98059, *** = *p* < 0.001, IRI vs. IRI + H_2_S, IRI + H_2_S vs. IRI + H_2_S + PD 98059 for HO-1, HSP-70, and SOD 1, ** = *p* < 0.01, IRI + H_2_S vs. IRI + H_2_S + PD 98059 for SOD-2 and Catalase).

## Data Availability

All data generated or analyzed during this study are included in this published article.

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
