# Peer review of "Inhalative as well as Intravenous Administration of H2S Provides Neuroprotection after Ischemia and Reperfusion Injury in the Rats’ Retina"

_ijms, 2022, doi:10.3390/ijms23105519_

Round 1
Reviewer 1 Report
This is a follow-up study investigating the protective effect of hydrogen sulfide in ischemia-reperfusion-induced retinal injury in rats. The authors present some new data and the experimental results are of interest.
I have several comments and suggestions:
- The statement that “the protective effect of H2S is more pronounced with inhalation therapy than intravenous administration” is somewhat misleading and requires more discussion. There is no doubt that it is meaningful to compare the effects of the two types of administration, however, the type of H2S (H2S and GYY4137), the mode of administration ((inhalation and intravenous administration),) and the difference in treatment time/effective concentration (I suppose) need to be considered. So the comparison and conclusion need to be limited to the conditions of this study.
- Please indicate whether the same group of rats was used for the WB/IHC assay and retinal ganglion cell count, since the time points of the tests were different (24h and 7days after IRI).
- Both female and male SD rats were used for this study. Whether there are any gender differences in the IRI animal model/ in the effect of H2S? Please comment on this.
- Fig 4, 9. It looks like expression of GFAP was increased in the retinal ganglion layer of IRI+H2S 1.5h group, and this was inhibited by PD inhibitor treatment. Please discuss.
In the Fig 4, 9 legends, description of arrows is missing.
- It is not clear why the authors performed pERK/GFAP double staining. Is pERK mainly expressed in astrocytes, and H2S exerts RGCs protection through astrocytes?
- In fig 2-8b there is one more + in column-4 and one less in column-6.
- Fig 11. Please give some indication that HO1, SOD, and catalase (antioxidant effects) are increased in the retina of the IRI model.
- In ref #17 by Biermann, H2S treatment inhibited pERK levels and thus protected RGCs. What are the possible reasons for this difference effect of H2S in the IRI model. Some discussion would be nice.
Author Response
Response to comments of Reviewer #1
We express our appreciation to the Reviewer for the time and effort spent in reviewing our work. We very much hope that the following responses and the revision of the manuscript have appropriately addressed the Reviewer´s comments.
Reviewer´s comments:
1)
Reviewer: The statement that “the protective effect of H2S is more pronounced with inhalation therapy than intravenous administration” is somewhat misleading and requires more discussion. There is no doubt that it is meaningful to compare the effects of the two types of administration, however, the type of H2S (H2S and GYY4137), the mode of administration ((inhalation and intravenous administration),) and the difference in treatment time/effective concentration (I suppose) need to be considered. So the comparison and conclusion need to be limited to the conditions of this study.
Reply: Thank you very much for this comment. We agree with you regarding this aspect, since we administered inhaled H2S at different times after the IR injury, but intravenous GYY was applied only at one dose and one time. In contrast to NaH2S, GYY releases H2S in a slow manner, so the timing of the single application is likely to play only a minor role in the context of postconditioning and has not been described to date. In one study, pre- and post-conditioning with GYY4137 was compared in murine intestinal ischemia-reperfusion injury and showed a better outcome (intestinal perfusion, histology) compared to preconditioning (Jensen et al., The route and timing of hydrogen sulfide therapy critically impacts recovery following ischemia and reperfusion injury, J Pediatr Surg, 2018). Regarding the dosage, the GYY4137 concentrations differ depending on the study performed, with most researchers choosing a higher dosage than the one used in our study. However, these doses are usually administered intraperitoneally (for example, renal IRI, 50 mg/kg i.p.: Zhao et al., Protective effect of GYY4137 on renal ischemia/reperfusion injury through Nrf-2 mediated antioxidant defence, Kidney Blood Press Res, 2021; ). An interesting study has investigated the effect of different doses of GYY4137 (12.5, 25, and 50 mg/kg) after myocardial ischemia-reperfusion injury and has reported the best effect (reduction of oxidative stress and apoptosis) using 50 mg/kg. However, in that study, rats were given GYY4137 intraperitoneally daily for seven days (Meng et al., GYY4137 protects against myocardial ischemia and reperfusion injury by attenuating oxidative stress and apoptosis in rats, J Biomed Res, 2015). Of course, we cannot exclude the possibility that a higher dosage would have been associated with a better neuroprotective effect in our study.
Therefore, we have now added the following limitations at the end of the discussion: “This study has some limitations to consider. In the present study, different time points were chosen for H2S administration, with the best effect observed at a time delay of 1.5 hours after the end of ischemia, whereas intravenous GYY4137 was administered only just before the end of ischemia. Although GYY4137 is a molecule that releases H2S slowly, the effect of a later application remains unclear. In addition, we administered only one concentration of GYY4137(25 mg/kg i.v.), and a higher dosage might be beneficial.”
2)
Reviewer: Please indicate whether the same group of rats was used for the WB/IHC assay and retinal ganglion cell count, since the time points of the tests were different (24h and 7days after IRI).
Reply: Thank you for this comment. These are different groups. As we described in the Methods section, the rats whose tissues were used for molecular analyses were enucleated after 24 h. Another group was used for retinal ganglion cell analysis. We chose the different time points to investigate the immediate molecular effect of hydrogen sulfide 24 h after IRI and to examine its long-term influence on retinal ganglion cells after 7 days.
3)
Reviewer: Both female and male SD rats were used for this study. Whether there are any gender differences in the IRI animal model/ in the effect of H2S? Please comment on this.
Reply: In fact, we have deliberately chosen to use both male and female animals in our research as a way to compensate for gender differences. The sex hormones have a well-known influence on outcomes after ischemic brain damage (Kim et al., Age and sex differences in the pathophysiology of acute CNS injury, Neurochemistry International, 2019). Nevertheless, we have also looked at the results from a gender-specific perspective, and we have found no differences in our in vivo model.
4)
Reviewer: Fig 4, 9. It looks like expression of GFAP was increased in the retinal ganglion layer of IRI+H2S 1.5h group, and this was inhibited by PD inhibitor treatment. Please discuss.
Reply: Thank you for this comment. Indeed, we saw increased GFAP expression after H2S treatment. As described in our manuscript, we assessed this as a transient Müller cell activation. Although gliosis is generally considered to represent a nonspecific response to retinal damage (compared to IRI), it can also serve a neuroprotective function in the early phase of injury and protect neuronal tissue from further damage. Since activation of extracellular signal-regulated kinases (ERK) also leads to a Müller cell response, increased GFAP expression is quite reasonable (Akiyama et al., Presence of mitogen-activated protein kinase in retinal Muller cells and its neuroprotective effect ischemia reperfusion injury, Neuroreport, 2002). This assumption may be strengthened by the reduction in GFAP levels after the application of the specific ERK1/2 inhibitor PD98059. Other neuroprotective properties of Müller cells include the regulation of water and ion homeostasis (Pannicke et al., Comparative electrophysiology of retinal Muller glial cells – a survey on vertebrate species, Glia, 2017; Tada et al., Inwardly rectifying K+ channel in retinal Muller cells: comparison with the KAB-2/Kir4.1 channel expressed in HEK293T cells, Jpn J Physiol, 1998) and the recycling of neurotransmitters, such as GABA and excitatory glutamate (Derouiche et al, Coincidence of L-glutamate/l-aspartate (GLAST) and glutamine synthase (GS) immunoreactions in retinal glia: evidence for coupling of GLAST and GS in transmitter clearance, J Neurosci Res, 1995), which could be stimulated by postconditioning with H2S. However, these investigations were not part of the present study.
5)
Reviewer: In the Fig 4, 9 legends, description of arrows is missing.
Reply: This has been corrected, as requested.
6)
Reviewer: It is not clear why the authors performed pERK/GFAP double staining. Is pERK mainly expressed in astrocytes, and H2S exerts RGCs protection through astrocytes?
Reply: Thank you for this comment. As we know, glial fibrillary acidic protein (GFAP) is a marker for glial cells, which consist of astrocytes, Müller cells, and microglia in the retina. In this study, we did not further differentiate the influence of H2S on the different retinal glial cells, which is actually a point of criticism. For this reason, we cannot give a concrete answer regarding which cell type is responsible in the context of H2S-mediated neuroprotection, as all three cell types could mediate neuroprotective effects in the context of retinal IRI (Cuenca et al., Cellular responses following retinal injuries and therapeutic approaches for neurodegenerative diseases, Prog Retin Eye Res, 2014). We can only speculate, based on our data, but we suspect that a reactive Müller cell activation is triggered by ERK phosphorylation. Nevertheless, we believe that investigations concerning retinal glia are important because of their contribution to the underlying pathophysiological processes, especially in the context of neuroinflammation. GFAP activation by H2S postconditioning is an important finding that requires further investigation and will be included in follow-up projects.
7)
Reviewer: In fig 2-8b there is one more + in column-4 and one less in column-6.
Reply: Thank you for pointing this out. We have fixed the error.
8)
Reviewer: Fig 11. Please give some indication that HO1, SOD, and catalase (antioxidant effects) are increased in the retina of the IRI model.
Reply: Thank you for this comment. We would like to explain the context: Stressors such as ischemia-reperfusion injury activate the heat shock response and elevate hemeoxgynase-1 (HO-1) levels. HO-1 is involved in defense mechanisms and helps to alleviate cellular damage. Our study shows that postconditioning with H2S reduced HO-1 expression after IRI. Hence, we conclude that H2S protects against cell stress. Free radicals (ROS) also play a critical role and are released in the context of retinal IRI. As a result, antioxidant enzymes, such as SOD and catalase, are activated, and these scavenge ROS and protect tissues against cytotoxicity. Because postconditioning with H2S reduces the activity of SOD and catalase, we hypothesize a reduced demand of antioxidant enzymes after IRI.
Reviewer 2 Report
The authors submitted a research article with the aim of elucidating the effects of either H2S inhalation or intravenous application of the H2S donor GYY 4137 on several intracellular signaling molecules that are key components of inflammatory and apoptotic pathways. They using animal model (Adult male and female Sprague-Dawley rats) to evaluate retinal ischemia/reperfusion injury and H2S treatment. The authors found that post-conditioning with 80 ppm H2S and intravenous GYY 4137 increased the phosphorylation of the MAP kinases p38 and ERK-1/2 that seems to be promoted an anti-apoptotic effect. Along with it, they noticed that post-conditioning with 80 ppm H2S and intravenous GYY 4137 reduced the expression of ICAM-1 and iNOS, which are key components of pro-inflammatory signaling cascades, and that treatment with the ERK pathway inhibitor PD 98059 suppressed the protective effect of H2S. These findings are appeared to be impressive and I congratulate the authors on them.Overall, the aim is clear and concise. The sections "Methods", "Results", and "Discussion" are well and thoroughly written and exhibit all aspects of the study. The tables and figures are clear and legible. The conclusive part is well structured. However, I would like to propose some issues to discuss.
- The authors should add a brief description of underlying molecular mechanisms of pre- and post- conditioning that involve the signal pathways mebtioned above.
- The conclusive part should be extended to give more comprehensive notion of clinical significance of the findigs.
- The authors should add a brief commentary toward pharmacology approach based on the results
Round 2
Reviewer 1 Report
Good job! I am satisfied with how the authors have addressed my comments.
A small comment on Question 6, I personally think that GFAP is a marker of glial cells, including astrocytes and Muller cells, and may not include microglia. Please consider.
Author Response
Thank you for your positive comment.
Probably the answer to question 6 was unclearly expressed. We only intended to state that microglia is one of three retinal glial cell types. Of course, we know that Iba-1 is a typical marker for retinal microglia, not GFAP.